**Data Availability Statement:** All relevant data are within the manuscript and its Supporting Information files.

# Current and future costs of cancer attributable to insufficient leisure-time physical activity in Brazil

**Ronaldo Corrêa Ferreira da Silva**[1‡], **Thainá Alves Malhão**📧[1☯], **Leandro F. M. Rezende**📧[2☯], **Rafael da Silva Barbosa**[3☯], **Arthur Orlando Correa Schilithz**[1‡], **Luciana Grucci Maya Moreira**[1‡], **Paula Aballo Nunes Machado**[1‡], **Fabio Fortunato Brasil de Carvalho**📧[1‡*], **Maria Eduarda Leão Diogenes**[1,4‡]

1 Cancer Prevention and Surveillance Coordination, Brazilian National Cancer Institute (INCA), Rio de Janeiro, Brazil, 2 Department of Preventive Medicine, Paulista School of Medicine, Federal University of São Paulo, São Paulo, Brazil, 3 Postgraduate Program in Social Policy, Federal University of Espírito Santo, Vitória, Espírito Santo, Brazil, 4 Basic and Experimental Nutrition Department, Nutrition Institute, State University of Rio de Janeiro, Rio de Janeiro, Brazil

☯ These authors contributed equally to this work.
‡ RCFS, AOCS, LGMM, PANM, FFBC and MELD also contributed equally to this work.
* fabio.carvalho@inca.gov.br

## Abstract

### Objectives

Cancer is an increasing cause of death and disability in Brazil and a pivotal vector for growing health expenditures. Lower levels of leisure-time physical activity are associated with a higher risk of some cancers. We quantified the current and future cancer direct healthcare costs attributable to insufficient leisure-time physical activity in Brazil.

### Methods

We performed a macrosimulation model using: (i) relative risks from meta-analyses; (ii) prevalence data of insufficient leisure-time physical activity in adults ≥ 20 years; (iii) national registries of healthcare costs of adults ≥ 30 years with cancer. We used simple linear regression to predict cancer costs as a function of time. We calculated the potential impact fraction (PIF) considering the theoretical-minimum-risk exposure and other counterfactual scenarios of physical activity prevalence.

### Results

We projected that the costs of breast, endometrial, and colorectal cancers may increase from US$ 630 million in 2018 to US$ 1.1 billion in 2030 and US$ 1.5 billion in 2040. The costs of cancer attributable to insufficient leisure-time physical activity may increase from US$ 43 million in 2018 to US$ 64 million in 2030. Increasing leisure-time physical activity could potentially save US$ 3 million to US$ 8.9 million in 2040 by reducing the prevalence of insufficient leisure-time physical activity in 2030.

**Funding:** This research received financial support from Climate and Land Use Alliance (CLUA) (Grant Number G-2007-56990) and Brazilian National Cancer Institute José Alencar Gomes da Silva (INCA). https://www.climateandlandusealliance.org/ www.inca.gov.br The funder had no role in study design, data collection and analysis, decision to publish, or preparation of the manuscript and does not necessarily share the positions expressed in the Grantee's publication.

**Competing interests:** The authors have declared that no competing interests exist.

## Conclusion

Our results may be helpful to guide cancer prevention policies and programs in Brazil.

## Introduction

Cancer is the second leading cause of death and disability-adjusted life years in Brazil [1]. Approximately 625,000 newly diagnosed cancer cases (450,000 excluding non-melanoma skin cancer cases) [2] and 224,000 cancer deaths (excluding non-melanoma cancer deaths) occurred per year in the country [3]. By 2040, projected estimates indicate that, due to population growth and aging, cancer cases and deaths may increase by 66% and 81%, respectively [3, 4].

Regular physical activity can prevent noncommunicable diseases (NCDs) such as heart disease, stroke, diabetes and some cancers [5]. Physical activity has been characterized and investigated in epidemiologic studies by the domains (e.g., occupational, leisure-time/recreational, household, and transport), volume (e.g., frequency, duration, and intensity), and lifetime period (ranging from current to lifetime activity).

More than 500 observational epidemiologic studies have examined some aspects of the association between physical activity and cancer incidence [6]. In 2018, the World Cancer Research Fund/American Institute for Cancer Research (WCRF/AICR) concluded that there is strong evidence that physical activity decreases the risk of breast (postmenopausal), colon, and endometrial cancer. Of note, most of the prospective cohort studies on physical activity and cancer included in the WCRF/AICR report assessed leisure-time physical activity in adults from high-income countries [7, 8].

The World Health Organization (WHO) recommends that adults do at least 150–300 minutes/week of moderate-intensity aerobic physical activity, 75–150 minutes/week of vigorous-intensity aerobic physical activity or an equivalent combination of moderate and vigorous physical activity [9].

Globally, the prevalence of insufficient physical activity (i.e., those not reaching the WHO recommendations) was 27.5% in 2016, with a higher prevalence in women (31.7%) than in men (23.4%) [10]. Progress towards achieving the global target of a 10% relative reduction of insufficient physical activity by 2025, described in the global action plan on the prevention and control of NCDs and endorsed to member states of WHO [5], has been too slow [10]. The prevalence of insufficient physical activity has been stable in the world between 2001 and 2016, although it shows a growth trend in high-income Western countries, Latin America and the Caribbean [10].

In Brazil, different surveys have measured population-level physical activity over the last few decades. Although they differ in study design, sampling, and how physical inactivity is characterized, they continue to provide important information for researchers and decision-makers. The prevalence of insufficient physical activity in the 2019 Brazilian National Health Survey varies according to the domain (occupational, leisure-time, household, and transport activity). The highest prevalence of insufficient physical activity occurs in the household tasks (84.2%) and leisure-time activities (69.9%), while the lowest was observed at work domain (57.4%). Women (47.5%), elders (59.7%), and individuals with no or low formal education (49.9%) have the highest prevalence of insufficient physical activity [11].

Some studies have quantified the impact of alternative scenarios of increasing population-wide physical activity on the burden of cancer in Brazil, both in terms of cancer cases and direct health care costs [12–14]. However, to our knowledge, studies quantifying the future economic burden of breast, colorectal and endometrium cancer attributable to insufficient

leisure-time physical activity in Brazil are lacking, despite its potential to inform public policies and prevention initiatives.

In this study, we estimated the current (2018) and future (2030 and 2040) federal direct healthcare costs of breast, colorectal and endometrial cancer in the Brazilian Public Health System (SUS) attributable to insufficient leisure-time physical activity. In addition, we estimated savings in federal direct healthcare costs of breast, colorectal, and endometrial cancer in 2040 through counterfactual scenarios of increasing population-wide leisure-time physical activity prevalence to be achieved in 2030.

## Materials and method

### Data and study design

We applied a top-down costing methodology and performed a macrosimulation model [15] to estimate the current (2018) and future (2030 and 2040) costs of cancer attributable to insufficient leisure-time physical activity in Brazil using the following data: 1.Relative risks (RR) from WCRF/AICR meta-analyses (**S1 File in S1 Appendix**); 2.Prevalence data (%) of insufficient leisure-time physical activity in adults aged 20 years or older who rely exclusively on the public health system from two nationally representative surveys; 3. Nationwide registries of federal direct healthcare costs of inpatient and outpatient procedures approved for payment in SUS regarding adults aged 30 years or older with cancer. A summary of the data considered in the macrosimulation model was presented in **S2 File in S1 Appendix**.

We estimated the potential impact of insufficient leisure-time physical activity on federal direct healthcare costs of cancer, assuming a 10-year time lag between exposition and outcome via comparative risk assessment [16]. We used the potential impact fraction (PIF) equation [17] and the Monte Carlo simulation method to estimate the PIF and its 95% uncertainty intervals, considering the theoretical-minimum-risk exposure and alternative counterfactual scenarios of the prevalence of insufficient leisure-time physical activity. We multiplied PIF by the direct healthcare costs of cancer to calculate the cancer costs attributable to insufficient leisure-time physical activity.

### Risk estimates and cancer sites

We included cancer sites with strong evidence of association (convincing or probable) with physical activity according to the WCRF/AICR: colon, endometrium, and breast (postmenopausal). We obtained the RR for the highest compared with the lowest level of leisure-time physical activity by sex (**Table 1**). To convert decreased risk associated with sufficient leisure-time physical activity to the increased risk associated with insufficient leisure-time physical activity we used the following equation:

$$RR = 1 + \log(\frac{1}{RR_x})$$

**Table 1. Relative risk (RR) and potential impact fraction (PIF) of insufficient leisure-time physical activity-associated cancers per sex.**

| Cancer type | Sex | Highest vs lowest RR (95% CI) | 2018 PIF % (95% UI) | 2030 PIF % (95% UI) |
|---|---|---|---|---|
| Breast (post-menopausal) | Women | 1.14 (1.06–1.21) | 11.60 (3.68–19.09) | 10.03 (2.84–16.85) |
| Colon | Women | 1.17 (1.09–1.25) | 13.99 (6.08–21.31) | 11.84 (5.04–18.28) |
| | Men | 1.17 (1.09–1.25) | 13.86 (6.14–20.93) | 10.91 (4.55–17.05) |
| Endometrium | Women | 1.31 (1.07–1.54) | 22.76 (1.63–39.44) | 19.40 (1.16–35.03) |

Abbreviations: RR, relative risk; PIF, potential impact fraction; CI, confidence intervals; UI, uncertainty interval.

where RR represents the increase in the risk associated with insufficient leisure-time physical activity. The RRx represents the decrease in the risk associated with sufficient leisure-time physical activity.

## Assessment of leisure-time physical activity in 2008 and 2019

We obtained data on leisure-time physical activity of 57,962 participants from the National Household Sample Survey (PNAD) 2008 [18] and 37,690 participants from the Brazilian National Health Survey (PNS) 2019 [11]. PNAD and PNS microdata are available in the public domain via the Brazilian Institute of Geography and Statistics (IBGE) in partnership with the Ministry of Health (**S3 File in S1 Appendix**). We considered only adults aged 20 years or older who reported not having health insurance to obtain the prevalence and 95% CI of leisure-time physical activity by sex. For postmenopausal breast cancer, we considered women aged 50 years or older. We incorporated the complex sample design into all estimates using the survey package in RStudio version 1.4.1103.

We considered insufficient leisure-time physical activity when the individual reached <7.5 metabolic equivalents of tasks (MET)-hours per week. To calculate the variable "MET-hours per week", we multiplied the weekly frequency of leisure-time physical activity by the duration in hours and the MET of the modality [19]. We described some methodological differences between the surveys in **S4 File in S1 Appendix**.

## Counterfactual scenarios for leisure-time physical activity

We proposed four counterfactual (alternative) scenarios of reductions in the prevalence of insufficient leisure-time physical activity observed in 2019 to be achieved in Brazil in 2030 to save direct healthcare costs with cancer in 2040: 1. Decreasing the prevalence of insufficient leisure-time physical activity by 10% [20]; 2. Increasing the prevalence of sufficient leisure-time physical activity by 30% [21]; 3. Increasing the practice of moderate leisure-time physical activity in the Brazilian population by 30 minutes in weekly volume, equivalent to 1.5 MET-hours per week [22]; 4. Reaching a 29.7% prevalence of sufficient leisure-time physical activity for both sexes, a value observed in men in 2019 [23]. We based the counterfactual scenarios on policy targets [20, 21] and theoretical discussion in the literature [22, 23].

## Federal direct healthcare costs of cancer in the Brazilian Unified Health System in 2018, 2030 and 2040

We retrieved registries of federal direct healthcare costs of inpatient and outpatient procedures related to cancer between 2008 and 2019 from the Hospital Information System (SIH) and Ambulatory Information System (SIA) of the SUS (**S3 File in S1 Appendix**). We used the 10[th] Revision of the International Statistical Classification of Diseases and Related Health Problems (ICD-10) codes for recovering cancer procedures from information systems (**S5 File in S1 Appendix**). We stratified the direct healthcare costs by sex and cancer type (all invasive cancers, breast cancer (any), postmenopausal breast cancer, colon cancer, colorectal cancer, and endometrial cancer). Assuming a 10-year time lag between exposure and outcomes, we considered the procedures approved for payment in adults with cancer aged 30 years or older in 2018, 2030, and 2040. For postmenopausal breast cancer, we considered women aged 60 years or older.

We performed a simple linear regression to predict the future costs of each cancer type evaluated (dependent variable) as a function of time (independent variable) up to 2030 and 2040 based on the historical values observed over time between 2008 and 2019. It is crucial to control for potential confounders while examining the possible determinants of cost. Since our outcome was the direct healthcare costs over time, it was unnecessary to control for

confounders because we observed their effect on the costs used to fit the regression model [24]. We transformed the monetary values of the Brazilian currency Real (R$) to United States Dollar (US$), considering the purchasing power parity (PPP) of 2018 (conversion factor 2.226) to current costs and 2019 (conversion factor 2.281) to future costs [25].

### Cancer costs attributable to insufficient leisure-time physical activity

Based on the abovementioned intermediate outputs of the models, we calculated the PIF for insufficient leisure-time physical activity-associated cancers by sex and counterfactual scenario using the following equation:

$$PIF = \frac{\sum_{i=1}^{n} P_i RR_i - \sum_{i=1}^{n} P'_i RR_i}{\sum_{i=1}^{n} P_i RR_i}$$

where P$i$ is the proportion of the population at the level $i$ leisure-time physical activity in a given year, P'$i$ is the proportion of the population at the level $i$ of leisure-time physical activity in a given counterfactual scenario, and RR$i$ is the relative risk of postmenopausal breast cancer, colon cancer, and endometrial cancer at the level $i$ of leisure-time physical activity. Levels $i$ for physical activity were <7.5 vs. ≥7.5 MET-hours per week. Of note, the PIF equals the Population Attributable Fraction (PAF) when the counterfactual scenario represents the theoretical minimum risk exposure level.

For colon and postmenopausal breast cancer, we recalculated the PIF by the topographic site: colorectal and breast (women aged 30 years or older), using the following equation:

$$PIF\ site = \frac{Attributable\ cost\ according\ to\ WCRF\ cancer\ type}{Total\ cost\ of\ topographic\ site}$$

In other words, we divided the colon and postmenopausal breast cancer costs attributable to insufficient physical activity by the total colorectal and breast (any) cancer cost, respectively.

We assessed the cancer costs attributable to insufficient leisure-time physical activity, multiplying PIF by the cancer costs. We considered the prevalence in 2008 and 2019 and the costs of cancer in 2018 and 2030, respectively. Finally, we calculated the potential costs saved in 2040 by increasing leisure-time physical activity to levels fixed in the counterfactual scenarios in 2030.

We quantified the uncertainty in all modeled estimates using the Monte Carlo simulation approach with 10,000 iterations. The simulation works thoroughly producing a draw from the distributions of a) baseline prevalence of leisure-time physical activity considering a binomial distribution, and b) the log of the RR for the association between insufficient leisure-time physical activity and cancer incidence considering a normal distribution. We calculated PIF by sex for the 50th, 2.5th, and 97.5th percentiles of estimates as the central estimate and 95% uncertainty intervals (UI) across all simulations. We used R Studio version 1.3.1093 for analysis.

This study was approved by the Institutional Review Board of the Brazilian National Cancer Institute José Alencar Gomes da Silva (INCA) at the Brazilian Ministry of Health (MS) (CAAE 12008119.8.0000.5274).

## Results

### Distribution of insufficient leisure-time physical activity in Brazil in 2008 and 2019

Comparing the years 2008 and 2019, prevalence of insufficient leisure-time physical activity (defined as <7.5 MET-h/week) declined from 92.3% to 70.3% in men and 94.2% to 76.9% in women (**Fig 1**).

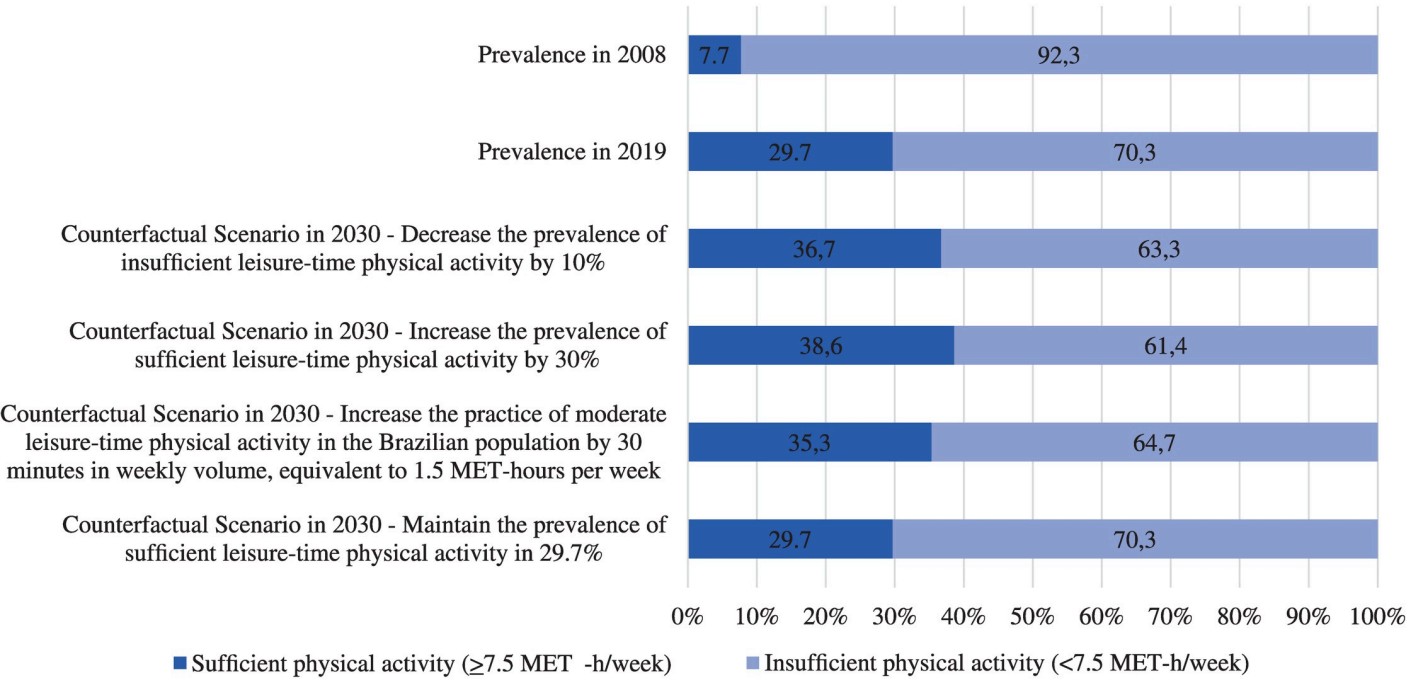

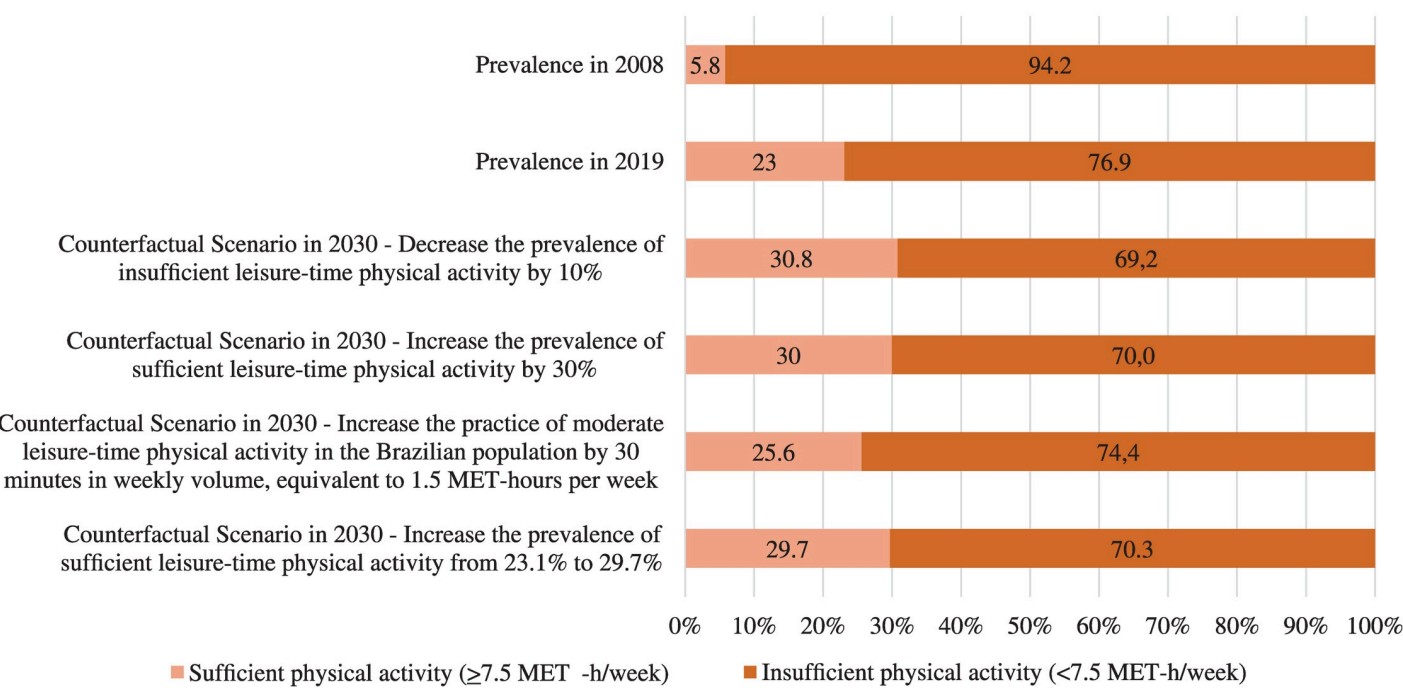

**Fig 1. Leisure-time physical activity in adults ≥ 20 years who rely exclusively on the public health system in Brazil in 2008 and 2019 and levels fixed in counterfactual scenarios to be achieved until 2030.**

## Current federal direct healthcare costs of breast, colorectal and endometrial cancer in Brazil in 2018 attributable to insufficient leisure-time physical activity

In 2018, federal direct healthcare costs of cancer were, approximately, US$ 1.6 billion, of which US$ 631 million were cancers-related to insufficient leisure-time physical activity. Cancer-related to insufficient leisure-time physical activity with highest costs was breast (US$ 366 million), followed by colorectal (US$ 245 million) and endometrium (US$ 20 million). Insufficient leisure-time physical activity associated cancers costs were higher in women (US$ 507 million) than in men (US$ 123 million).

We estimated that 2.7% (US$ 43 million) federal direct healthcare costs of cancer were attributable to insufficient leisure-time physical activity. Cancers with the highest PIF were endometrium (22.8%), followed by colorectal (8.7%) and breast (4.5%). Cancer with the highest attributable costs was colorectal (US$ 21 million), followed by breast (US$ 17 million) and endometrium (US$ 5 million) (**Table 2**).

## Future federal direct healthcare costs of breast, colorectal and endometrial cancer in Brazil in 2030 attributable to insufficient leisure-time physical activity

In 2030, we projected an approximately US$ 2.6 billion in direct healthcare costs of cancer in Brazil, of which US$ 1.1 billion will be cancers related to insufficient leisure-time physical

**Table 2. Current (2018) and future (2030) federal direct healthcare costs (US$ Million) of cancer attributable to insufficient leisure-time physical activity in Brazil.**

| Cancer type | Sex | 2018 | | | 2030 | | |
|---|---|---|---|---|---|---|---|
| | | PIF % (95% UI) | Cancers costs* US$ Million | Attributable costs US$ Million (95% UI) | PIF % (95% UI) | Cancers costs** US$ Million | Attributable costs US$ Million (95% UI) |
| Breast (any) | Women | 4.52 (1.44–7.44) | 365.63 | 16.54 (5.25–27.21) | 3.97 (1.13–6.67) | 620.39 | 24.64 (6.98–41.37) |
| Colorectal | Women | 9.12 (3.96–13.89) | 121.23 | 11.06 (4.80–16.84) | 7.83 (3.33–12.08) | 220.93 | 17.29 (7.36–26.69) |
| | Men | 8.36 (3.70–12.63) | 123.70 | 10.35 (4.58–15.62) | 6.72 (2.80–10.50) | 228.91 | 15.39 (6.41–24.04) |
| | Total | 8.74 (3.83–13.25) | 244.93 | 21.40 (9.38–32.46) | 7.26 (3.06–11.28) | 449.84 | 32.68 (13.77–50.72) |
| Endometrium | Women | 22.76 (1.63–39.44) | 20.14 | 4.58 (0.33–7.94) | 19.40 (1.16–35.03) | 36.62 | 7.10 (0.43–12.83) |
| Insufficient leisure-time physical activity -associated cancer | Women | 6.35 (2.05–10.25) | 507.00 | 32.18 (10.38–51.99) | 5.59 (1.68–9.21) | 877.94 | 49.03 (14.77–80.89) |
| | Men | 8.36 (3.70–12.63) | 123.70 | 10.35 (4.58–15.62) | 6.72 (2.80–10.50) | 228.91 | 15.39 (6.41–24.04) |
| | Total | 6.74 (2.37–10.72) | 630.70 | 42.52 (14.96–67.61) | 5.82 (1.91–9.48) | 1,106.85 | 64.42 (21.18–104.92) |
| All invasive cancers*** | Women | 3.66 (1.18–5.91) | 880.03 | 32.18 (10.38–51.99) | 3.38 (1.02–5.58) | 1,450.00 | 49.03 (14.77–80.89) |
| | Men | 1.49 (0.66–2.24) | 696.19 | 10.35 (4.58–15.62) | 1.35 (0.56–2.11) | 1,137.20 | 15.39 (6.41–24.04) |
| | Total | 2.70 (0.95–4.29) | 1,576.22 | 42.52 (14.96–67.61) | 2.49 (0.82–4.06) | 2,587.20 | 64.42 (21.18–104.92) |

Abbreviations: PIF, potential impact fraction, considering the theoretical-minimum-risk exposure; UI, uncertainty interval; *Observed direct healthcare costs (inpatient and outpatients) of cancer in adults ≥30 years. **Projected direct healthcare costs (inpatient and outpatients) of cancer in adults ≥30 years. ***Codes of 10[th] revision of the International Statistical Classification of Diseases and Related Health Problems (ICD-10): C00-C97.

**Table 3. The potential impact of leisure-time physical activity counterfactual scenarios in 2030 on reduction of projected federal direct healthcare costs of cancer in Brazil in 2040 (US$ Million).**

| Cancer type | Sex | Projected cancers costs in 2040 * | Counterfactual Scenario 1 | Counterfactual Scenario 2 | Counterfactual Scenario 3 | Counterfactual Scenario 4 |
|---|---|---|---|---|---|---|
| | | | US$ Million (95% UI) | | | |
| Breast (any) | Women | 834.70 (763,84–905,55) | 3.36 (0.00–7.28) | 2.39 (0.00–6.11) | 0.69 (0.00–4.10) | 0.17 (0.00–3.62) |
| Colorectal | Women | 305.06 (270.74–339.38) | 2.43 (0.00–4.95) | 2.16 (0.00–4.66) | 0.78 (0.00–3.06) | 2.04 (0.00–4.54) |
| | Men | 317.43 (283.18–351.68) | 2.17 (0.00–4.78) | 2.72 (0.00–5.44) | 1.72 (0.00–4.24) | 0.00 (0.00–2.34) |
| | Total | 622.49 (553.92–691.06) | 4.60 (0.00–9.73) | 4.88 (0.00–10.10) | 2.50 (0.00–7.30) | 2.04 (0.00–6.88) |
| Endometrium | Women | 50.08 (43.35–56.80) | 0.97 (0.00–1.91) | 0.87 (0.00–1.74) | 0.31 (0.00–0.92) | 0.85 (0.00–1.69) |
| Insufficient leisure-time physical activity -associated cancer | Women | 1,189.84 (1,077.94–1,301.74) | 6.76 (0.00–14.15) | 5.42 (0.00–12.51) | 1.78 (0.00–8.08) | 3.06 (0.00–9.86) |
| | Men | 317.43 (283.18–351.68) | 2.17 (0.00–4.78) | 2.72 (0.00–5.44) | 1.72 (0.00–4.24) | 0.00 (0.00–2.34) |
| | Total | 1,507.27 (1,361.12–1,653.42) | 8.93 (0.00–18.93) | 8.14 (0.00–17.95) | 3.50 (0.00–12.32) | 3.06 (0.00–12.20) |
| All invasive cancers** | Women | 1,930.44 (1,743.55–2,117.32) | 6.76 (0.00–14.15) | 5.42 (0.00–12.51) | 1.78 (0.00–8.08) | 3.06 (0.00–9.86) |
| | Men | 1,506.06 (1,324.66–1,687.46) | 2.17 (0.00–4.78) | 2.72 (0.00–5.44) | 1.72 (0.00–4.24) | 0.00 (0.00–2.34) |
| | Total | 3,436.50 (3,068.20–3,804.79) | 8.93 (0.00–18.93) | 8.14 (0.00–17.95) | 3.50 (0.00–12.32) | 3.06 (0.00–12.20) |

Abbreviations: UI, uncertainty interval; Notes: Assuming that increasing sufficient leisure-time physical activity cannot harm, we replaced the negative values with zero. *Projected direct healthcare costs (inpatient and outpatients) of cancer in adults ≥30 years. **Codes of 10th revision of the International Statistical Classification of Diseases and Related Health Problems (ICD-10): C00-C97. Scenario 1: Decrease the prevalence of insufficient leisure-time physical activity by 10%; Scenario 2: Increase the prevalence of sufficient leisure-time physical activity by 30%; Scenario 3: Increase the practice of moderate leisure-time physical activity in the Brazilian population by 30 minutes in weekly volume, equivalent to 1.5 MET-hours per week; Scenario 4: Reach a 29.7% prevalence of sufficient leisure-time physical activity for both sexes, value observed in men in 2019.

activity. We estimated that nearly 2.5% of federal direct healthcare costs in 2030 (US$ 64 million) may be attributable to insufficient leisure-time physical activity. Women will contribute with US$ 49 million and men with US$ 15 million (**Table 2**).

### The potential impact of the reduction in the prevalence of insufficient leisure-time physical activity on projected direct healthcare costs of breast, colorectal and endometrial cancer in Brazil in 2040

We showed in Fig 1 four counterfactual scenarios of insufficient leisure-time physical activity in Brazil to be achieved in 2030. Projections of direct federal costs on cancer in 2040 are presented in Table 3. In 2040, approximately US$ 3.4 billion of federal direct healthcare costs in Brazil may be spent with cancer, which US$ 1.5 billion will be costs with cancers associated to insufficient physical activity.

We estimated that around US$ 3 to 8.9 million in 2040 could be saved by increasing population-wide leisure-time physical activity until 2030 (**Table 3**). The counterfactual scenario with highest potential impact on cancer costs was the 10% decrease in the prevalence of insufficient leisure-time physical activity (US$ 8.9 million).

## Discussion

Our study estimated that insufficient leisure-time physical activity was responsible for US\$ 43 million of direct cancer healthcare costs in 2018. Direct healthcare costs attributable to insufficient leisure-time physical activity may increase to US\$ 64 million in 2030. We also estimated that approximately US\$ 3 to 8.9 million in 2040 could be saved by increasing sufficient leisure-time physical activity or decreasing insufficient leisure-time physical activity at the population level.

The 2019 Brazilian National Health Survey in Brazil [11] revealed that physical activity showed differences between sexes during the leisure-time domain. Women (26%) are less physically active than men (34%), and in both sexes, physical activity increases in proportion to the increase in the level of formal education and decreases with age. We analyzed data on leisure-time physical activity of adults aged 20 years or older who reported not having health insurance from the same survey. Our data showed that 29.7% of men were physically active compared to 23.1% of women, which is expected since this is a lower-income population dependent on the public health system.

Insufficient physical activity has been associated with many chronic diseases and early deaths and poses a societal challenge due to the economic burden on countries, health systems, and families [26]. Around Int\$ 1.6 billion of direct costs, counting the five major NCDs (coronary heart disease, stroke, type 2 diabetes, breast cancer, and colon cancer), was attributable to insufficient physical activity in Brazil in 2013, with 48.2% of costs paid by the public sector [26]. A recent study conducted in Brazil found that insufficient physical activity accounted for Int\$ 50.3 million in direct healthcare costs for colon and post-menopausal breast cancers in 2017 [12]. We estimated that US\$ 43 million federal in direct healthcare costs of cancer in 2018 were attributable to insufficient leisure-time physical activity, with the highest attributable costs to colorectal (US\$ 21 million), followed by breast (US\$ 17 million) and endometrium (US\$ 5 million). The differences in values found in the data from Brazil between the studies are expected since the authors used different data on prevalence, RR, types of cancers, and financial year of reference. Our study adds information to previously study providing direct costs of endometrial cancer attributed to insufficient leisure-time physical activity, in addition to breast and colon cancers. It is well recognized that physical activity reduces the risk of endometrial cancer [7, 8, 27, 28]. Moreover, different from the previous study, we considered a time interval of 10 years between exposure and the onset of the disease. In addition, we provided the future cost of cancer attributable to insufficient leisure-time physical activity in Brazil considering four alternative counterfactual scenarios based on policy targets and theoretical discussion in the literature. Finally, we considered only leisure-time physical activity in our estimates, as we argue that reporting cancer treatment costs attributable to this domain can be useful to guide decision-makers in funding public policies and programs.

Considering the prevalence of insufficient leisure-time physical activity in 2019, and the projected economic burden of colorectal, breast, and endometrial cancer in Brazil, we estimated that insufficient leisure-time physical activity may be responsible for nearly 2.5% of direct cancer costs (US\$ 64 million) in 2030, a little more than a 50% increase from 2018 values in just over a decade. Women will account for 76% (US\$ 49 million) of direct cancer costs attributable to insufficient leisure-time physical activity in 2030. Other studies also found a higher relative share of women in cancer cases (and probably costs) associated with physical inactivity in other countries [23, 29].

Increasing physical activity to counterfactual scenarios in 2030 could save around US\$ 3 to 8.9 million in direct healthcare costs in 2040. The counterfactual scenarios sought to measure the economic impact of meeting national and international goals, correcting the difference in

prevalence between sexes and promoting small increments in the time of physical activity per week. The scenarios (scenarios 1 and 2) that met the national and global targets [20, 21] showed the most significant reductions in cancer costs associated with insufficient physical activity. The scenarios (scenarios 3 and 4) that correspond to small weekly increments in physical activity and those that seek to equalize physical activity levels between sexes [22, 23] had the lowest economic impact.

These findings could be explained by the principle of prevention that was introduced by Geoffrey Rose in the 1980s. This principle states that reducing a risk factor by a small amount in the general population instead of selectively reducing it by a large amount in high-risk individuals prevents a greater number of diseases. This is called a population-based prevention strategy [30]. An analysis of the World Health Organization´s Global Burden of Disease database supported that many major global risks are widely spread in a population, rather than restricted to a minority. Population-based strategies that seek to shift the whole distribution of risk factors often have the potential to produce substantial reductions in disease burden [31].

Significant increases in cancer incidence and mortality are expected to occur in Brazil, where economic resources for cancer care are limited. Primary preventive strategies for cancer risk reduction could help to decrease the cancer burden. Given that almost two-thirds of adults are physically inactive in their leisure-time [11], a domain of physical activity that public policies may help to improve, it looks promising investing in physical activity to reduce the risk of cancer development and economic burden.

Evidence shows that many effective interventions (on the environment, mass media, the community and primary health care) can be implemented by policy-makers to increase people's physical activity in leisure-time. Multicomponent interventions adapted to local cultural and environmental contexts are the most successful and should use the existing social structures and participation of all stakeholders to reduce barriers to implementation [32]. In Brazil, local and federal governments have implemented some policies, programs and actions to encourage and increase physical activity. A scoping review recently evaluated community physical activity programs and found positive impacts on users' health indicators and leisure-time physical activity [33] and, at the end of 2021, the Ministry of Health launched the first Physical Activity Guide for the Brazilian Population [34].

The WCRF designed the MOVING framework, a tool to help policymakers, researchers and civil society organizations worldwide act to increase levels of physical activity. The framework is comprised of four policy domains: active societies, active environments, active people and active systems. The WCRF also offers the MOVING database that is a repository of global data on physical activity policy actions, containing information on what governments around the world have implemented to encourage people to be physically active. The database collects policy actions from around the world which are implemented on a national level, and are currently in effect [35].

Our study has some limitations. The surveys included in the study used different methodologies to classify the practice of physical activity, which limits their comparability. We included in our estimates only cancer sites with strong evidence by WCRF/AICR. These criteria may have underestimated the overall contribution of physical activity to cancer prevention considering that there is some evidence of associations with other cancer sites [27, 36]. Estimates of RR for the association between physical activity and cancer in the Brazilian population are inexistent, so we used RR derived from a meta-analysis using data from cohort studies conducted mainly in US and European countries. Some, but not all, studies have adjusted BMI and other potential factors that are difficult to define whether they act as confounders or mediators of the association between physical activity and cancer. Finally, future modeling

approaches using e.g., proportional multistate lifetable modeling of preventive interventions may be helpful to evaluating the long-term impacts of preventive interventions [37].

## Conclusion

Insufficient leisure-time physical activity was estimated to be responsible for US$ 43 million in direct healthcare costs of colorectal, breast, and endometrial cancers in 2018. In 2030, attributable direct healthcare costs may increase to US$ 64 million. We estimated that, approximately, US$ 3 to 8.9 million in 2040 could be saved by increasing physical activity at the population level. Our results may be helpful to guide decision-makers in funding cancer prevention policies and programs through physical activity in Brazil to seek an adequate balance between expenses on prevention by increasing physical activity at the population level and treatment of cancer.

## Supporting information

**S1 Appendix.**
(DOC)

## Author Contributions

**Conceptualization:** Ronaldo Corrêa Ferreira da Silva, Thainá Alves Malhão, Maria Eduarda Leão Diogenes.

**Formal analysis:** Thainá Alves Malhão, Leandro F. M. Rezende, Rafael da Silva Barbosa, Arthur Orlando Correa Schilithz.

**Funding acquisition:** Maria Eduarda Leão Diogenes.

**Methodology:** Ronaldo Corrêa Ferreira da Silva, Thainá Alves Malhão, Leandro F. M. Rezende, Rafael da Silva Barbosa, Arthur Orlando Correa Schilithz, Luciana Grucci Maya Moreira.

**Writing – original draft:** Ronaldo Corrêa Ferreira da Silva.

**Writing – review & editing:** Ronaldo Corrêa Ferreira da Silva, Thainá Alves Malhão, Leandro F. M. Rezende, Rafael da Silva Barbosa, Arthur Orlando Correa Schilithz, Luciana Grucci Maya Moreira, Paula Aballo Nunes Machado, Fabio Fortunato Brasil de Carvalho, Maria Eduarda Leão Diogenes.

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
