## [Decision Letter · Decision Letter 0]

16 Aug 2022

PONE-D-22-05640Current and future costs of cancer attributable to insufficient leisure-time physical activity in Brazil.PLOS ONE

Dear Dr. Carvalho,

Thank you for submitting your manuscript to PLOS ONE. After careful consideration, we feel that it has merit but does not fully meet PLOS ONE’s publication criteria as it currently stands. Therefore, we invite you to submit a revised version of the manuscript that addresses the points raised during the review process.

 Two reviewers have identified a number of concerns that need to be carefully addressed in your revision. Please pay particular attention to the methodological clarifications the reviewers have requested, to ensure that your manuscript satisfied our third publication criterion.

We look forward to receiving your revised manuscript.

Kind regards,

Jamie Males

Editorial Office

PLOS ONE

Journal Requirements:

“This research received financial support from Climate and Land Use Alliance (CLUA) (Grant Number G-2007-56990) and Brazilian National Cancer Institute José Alencar Gomes da Silva (INCA).

https://www.climateandlandusealliance.org/

www.inca.gov.br

The funder had no role in study design, data collection and analysis, decision to publish, or preparation of the manuscript and does not necessarily share the positions expressed in the Grantee's publication.”

Reviewers' comments:

Reviewer's Responses to Questions

**Comments to the Author**

1. Is the manuscript technically sound, and do the data support the conclusions?

Reviewer #1: Partly

Reviewer #2: Yes

2. Has the statistical analysis been performed appropriately and rigorously? 

Reviewer #1: Yes

Reviewer #2: Yes

3. Have the authors made all data underlying the findings in their manuscript fully available?

Reviewer #1: Yes

Reviewer #2: Yes

4. Is the manuscript presented in an intelligible fashion and written in standard English?

Reviewer #1: Yes

Reviewer #2: Yes

5. Review Comments to the Author

Reviewer #1: The paper seeks to address important questions but the authors need to put a major effort into explaining their methodology. For example, what exactly is the “top-down macrosimulation” that authors performed? What is the “comparative risk assessment”?What is the PIF equation? Neither these foundational concepts are adequately referenced nor explained in detail. This paper needs major econometric evaluation and until they are provided, I cannot comment on the further merit of the paper. Please provide every equation that is being used in the paper.

Reviewer #2: This is well conducted piece of research, which addresses an important topic for public health policy and practice. I have few substantial suggestions to make.

Introduction

Page 3, Paragraph 1. Error in this section. The authors write “… 625,000 newly diagnosed cancer cases (450,000 excluding non-melanoma skin cancer cases) and 450,000 cancer deaths ..” It is not probable that there were the same number of new diagnoses as deaths.

Page 3, Paragraph 2. Suggest changing “Regular physical activity is proven to prevent …” to “Regular physical activity can prevent …”

Page 3, Paragraph 3. It’s not clear why the authors reference and discuss the PAGAC report when they do not utilize the conclusions of this report for the purpose of their analyses. It is fine to acknowledge that you used the WCRF Third Expert Report to inform decisions about cancer sites to include in the analyses – but reference to the PAGAC report is a bit out of place in the Introduction. Makes more sense to bring this reference in in the Discussion (when you are talking about potential underestimation of cases, costs etc).

Page 4, Paragraph 2. The second sentence seems out of context “Presse towards achieving the global target of a 10% relative reduction of insufficient physical activity by 2025 has been too slow” – need to know the reference, who set this target?

Page 4, Paragraph 3. Do not need the “the” in the first sentence … “measuring the physical activity over …”

Page 5, Paragraph 2. Change “In addition, we estimated saves in federal direct ..” to “In addition, we estimated savings in federal direct …”

Methods

Page 5, Paragraph 3. Are the RRs presented in S1 File adjusted for BMI or other measures of body composition? It is always difficult (when dealing with single exposure measures) to understand whether BMI is a confounder or mediator of the relationship between physical activity and cancer.

Page 6. Table 1. Just to clarify – the PIFs provided in Table 1 represent the highest vs lowest physical activity categories (which is essentially the PAF)?

Discussion

Page 17, Paragraph 2. Suggest authors use full numbers (no decimals) throughout text. There is a mix in the paragraph.

Page 17, Paragraph 3. The authors note that another recent study in Brazil has estimated the costs associated insufficient physical activity. It’s fine that others have done similar work – but the authors really need to make the case for what their research adds. Why are the inputs (RRs, populations estimates) more robust that prior studies?

Page 18, Paragraph 2. The authors present the cost of physical inactivity in terms of cancer treatment; but what is the estimated cost for increasing physical activity at the population level? Running public health programmes also comes with a cost.

6. PLOS authors have the option to publish the peer review history of their article (what does this mean?). If published, this will include your full peer review and any attached files.

Reviewer #1: No

Reviewer #2: No

---

## [Author Response · Author response to Decision Letter 0]

31 Aug 2022

Reviewer #1: The paper seeks to address important questions but the authors need to put a major effort into explaining their methodology. For example, what exactly is the “top-down macrosimulation” that authors performed? What is the “comparative risk assessment”? What is the PIF equation? Neither these foundational concepts are adequately referenced nor explained in detail. This paper needs major econometric evaluation and until they are provided, I cannot comment on the further merit of the paper. Please provide every equation that is being used in the paper.

R: Thanks for these considerations. We agree that although the terms are commonly used in epidemiologic and economic modelling studies (e.g., The Global Burden of Disease Study), the fundamental concepts of these expressions need to be better referenced. Thus, we have included 3 references that bring the fundamental concepts of expressions: “top-down approach” (lines 98-99), “comparative risk assessment” (line 109) and “PIF equation” (line 109). Finally, the PIF equation was explained in line 194. 

Reviewer #2: This is well conducted piece of research, which addresses an important topic for public health policy and practice. I have few substantial suggestions to make.

R: Thank you for your positive comments and careful reading.

Introduction

Page 3, Paragraph 1. Error in this section. The authors write “… 625,000 newly diagnosed cancer cases (450,000 excluding non-melanoma skin cancer cases) and 450,000 cancer deaths ..” It is not probable that there were the same number of new diagnoses as deaths.

R: Corrected as suggested (Line 46). 

Page 3, Paragraph 2. Suggest changing “Regular physical activity is proven to prevent …” to “Regular physical activity can prevent …”

R: Corrected as suggested (Line 49).

Page 3, Paragraph 3. It’s not clear why the authors reference and discuss the PAGAC report when they do not utilize the conclusions of this report for the purpose of their analyses. It is fine to acknowledge that you used the WCRF Third Expert Report to inform decisions about cancer sites to include in the analyses – but reference to the PAGAC report is a bit out of place in the Introduction. Makes more sense to bring this reference in in the Discussion (when you are talking about potential underestimation of cases, costs etc).

R: Done as suggested (Lines 57, 317).

Page 4, Paragraph 2. The second sentence seems out of context “Presse towards achieving the global target of a 10% relative reduction of insufficient physical activity by 2025 has been too slow” – need to know the reference, who set this target?

R: Done as suggested (Lines 69).

Page 4, Paragraph 3. Do not need the “the” in the first sentence … “measuring the physical activity over …”

R: Corrected as suggested (Line 73).

Page 5, Paragraph 2. Change “In addition, we estimated saves in federal direct ..” to “In addition, we estimated savings in federal direct …”

R: Corrected as suggested (Line 91).

Methods

Page 5, Paragraph 3. Are the RRs presented in S1 File adjusted for BMI or other measures of body composition? It is always difficult (when dealing with single exposure measures) to understand whether BMI is a confounder or mediator of the relationship between physical activity and cancer.

R: Thank you for this comment. We retrieved RR from the WCRF meta-analysis, which showed heterogeneity between studies in regard to adjustment by confounding factors, including BMI. Some, but not all studies adjusted by BMI in their multivariable models. We have now highlighted this as a potential limitation of our study (Lines 377-379). 

Page 6. Table 1. Just to clarify – the PIFs provided in Table 1 represent the highest vs lowest physical activity categories (which is essentially the PAF)?

R: In table 1 we present the 2018 PIF% and also the 2030 PIF%. The 2018 PIF represents the theoretical minimum risk scenario. Therefore, in this condition, PIF is equal to PAF as described in lines 199-201. On the other hand, we consider different contractual scenarios considering the mitigation of risk factors to calculate PIF 2030. Therefore, PIF 2030 is not essentially the PAF because did not consider the elimination of the risk factor (theoretical minimum risk exposure levels).

Discussion

Page 17, Paragraph 2. Suggest authors use full numbers (no decimals) throughout text. There is a mix in the paragraph.

R: : Corrected as suggested (Lines 38, 272, 292, 331-332, 388)

Page 17, Paragraph 3. The authors note that another recent study in Brazil has estimated the costs associated insufficient physical activity. It’s fine that others have done similar work – but the authors really need to make the case for what their research adds. Why are the inputs (RRs, populations estimates) more robust that prior studies?

R: It is recognized that physical activity reduces the risk of endometrial cancer. Our study adds information providing direct costs of endometrial cancer attributed to insufficient leisure-time physical activity, in addition to breast and colon cancers. Moreover, different to the previous study, we considered a time interval of 10 years between exposure and the onset of the disease. In addition, we provided the future cost of cancer attributable to insufficient leisure-time physical activity in Brazil considering four alternative counterfactual scenarios based on policy targets and theoretical discussion in the literature. Finally, we considered only leisure-time physical activity, while the previous study considered total physical activity. We argue that reporting cancer treatment costs attributable leisure-time physical activity can be useful to guide decision makers in funding public policies and programs to promote leisure-time physical activity. We explain what our research adds in lines 314-323.

Page 18, Paragraph 2. The authors present the cost of physical inactivity in terms of cancer treatment; but what is the estimated cost for increasing physical activity at the population level? Running public health programmes also comes with a cost.

R: The estimated cost for increasing physical activity at the population level was not the objective of this study. However, we agree that it is important to mentioned this point (Lines 390 - 392)

---

## [Decision Letter · Decision Letter 1]

2 Jun 2023

Current and future costs of cancer attributable to insufficient leisure-time physical activity in Brazil.

PONE-D-22-05640R1

Dear Dr. de Carvalho,

We’re pleased to inform you that your manuscript has been judged scientifically suitable for publication and will be formally accepted for publication once it meets all outstanding technical requirements.

Note from PLOS Staff: Please note that the comments from Reviewer 3 should be disregarded as they refer to a different manuscript and were submitted by the reviewer in error. Please accept our apologies for this and for not resolving this issue sooner.

Kind regards,

Felicity Hey

Editorial Research Associate, PLOS

On behalf of,

Vitor Barreto Paravidino

Academic Editor

PLOS ONE

Additional Editor Comments (optional):

Reviewers' comments:

Reviewer's Responses to Questions

**Comments to the Author**

1. If the authors have adequately addressed your comments raised in a previous round of review and you feel that this manuscript is now acceptable for publication, you may indicate that here to bypass the “Comments to the Author” section, enter your conflict of interest statement in the “Confidential to Editor” section, and submit your "Accept" recommendation.

Reviewer #2: All comments have been addressed

Reviewer #3: All comments have been addressed

2. Is the manuscript technically sound, and do the data support the conclusions?

Reviewer #2: Yes

Reviewer #3: Partly

3. Has the statistical analysis been performed appropriately and rigorously? 

Reviewer #2: Yes

Reviewer #3: No

4. Have the authors made all data underlying the findings in their manuscript fully available?

Reviewer #2: Yes

Reviewer #3: No

5. Is the manuscript presented in an intelligible fashion and written in standard English?

Reviewer #2: Yes

Reviewer #3: Yes

6. Review Comments to the Author

Reviewer #2: The authors have addressed all suggested changes and queries appropriately. Thank you for the opportunity to review this work.

Reviewer #3: The article presents an important object of study, breast cancer prevention. However, some important articles do not make it readable for publication.

1- Breast self-examination as a breast revision prevention strategy has not been recommended in many systematic studies with analysis, "Breast self-examination as a prevention strategy in relation to the breast has not been recommended in many systematic studies with analysis , "Encounter with breast prevention, with regard to the reduction of overall mortality and analysis of breast cancer in women between 31 and 64 years" (KÖSTERS, J. P., GØTZSCHE, P. C. Regular self-examination or clinical exam for early detect of Breast Cancer. The Cochrane database of Systematics Reviews, Oxford, n. 2, 2013) The authors do not have this discussion in the manuscript, have there been changes in the changes in the teaching and performance of self-examination as a preventive measure?

2- Is there an attention network to assist women who will identify solutions? If so, it would be important to introduce the reader

3- What is the epidemiological situation of breast cancer in Syria?

4- Methodology

- Better describe the knowledge of breast self-examination. How will the degree of knowledge be assessed?

- Better describe how descriptive and bivariate statistical analyzes were performed (which statistical tests were used and why)

Results

They deserve to be better structured. I suggest that the results presented in figures be shown in tables, in which the p value and the statistical tests used will be presented. The figures are in poor resolution, the X and Y axes do not have a legend.

7. PLOS authors have the option to publish the peer review history of their article (what does this mean?). If published, this will include your full peer review and any attached files.

Reviewer #2: No

Reviewer #3: No

---

## [Editor Report · Acceptance letter]

30 Jun 2023

PONE-D-22-05640R1 

Current and future costs of cancer attributable to insufficient leisure-time physical activity in Brazil 

Dear Dr. Carvalho:

I'm pleased to inform you that your manuscript has been deemed suitable for publication in PLOS ONE. Congratulations! Your manuscript is now with our production department. 

Kind regards, 

on behalf of

Dr. Vitor Barreto Paravidino 

Academic Editor

PLOS ONE